# Influence of Nutritional Status and Physical Exercise on Immune Response in Metabolic Syndrome

**DOI:** 10.3390/nu14102054

**Published:** 2022-05-13

**Authors:** Mauro Lombardo, Alessandra Feraco, Chiara Bellia, Luigi Prisco, Ilenia D’Ippolito, Elvira Padua, Maximilian Andreas Storz, Davide Lauro, Massimiliano Caprio, Alfonso Bellia

**Affiliations:** 1Department of Human Sciences and Promotion of the Quality of Life, San Raffaele Roma Open University, 00166 Rome, Italy; alessandra.feraco@uniroma5.it (A.F.); luigi.prisco@uniroma5.it (L.P.); elvira.padua@uniroma5.it (E.P.); massimiliano.caprio@uniroma5.it (M.C.); bellia@med.uniroma2.it (A.B.); 2Laboratory of Cardiovascular Endocrinology, IRCCS San Raffaele Roma, 00166 Rome, Italy; 3Department of Biomedicine, Neurosciences, and Advanced Diagnostics, University of Palermo, 90127 Palermo, Italy; chiara.bellia@unipa.it; 4Department of Systems Medicine, University of Rome “Tor Vergata”, 00133 Rome, Italy; ileniadippolito@gmail.com (I.D.); d.lauro@med.uniroma2.it (D.L.); 5School of Human Movement Science, University of Rome “Tor Vergata”, 00133 Rome, Italy; 6Department of Internal Medicine II, Center for Complementary Medicine, Faculty of Medicine, University of Freiburg, 79106 Freiburg, Germany; maximilian.storz@uniklinik-freiburg.de

**Keywords:** nutrition, physical activity, immune function, aging, skeletal muscle, adipose tissue, adipokines, obesity, COVID-19, myokines

## Abstract

Metabolic Syndrome (MetS) is a cluster of metabolic alterations mostly related to visceral adiposity, which in turn promotes glucose intolerance and a chronic systemic inflammatory state, characterized by immune cell infiltration. Such immune system activation increases the risk of severe disease subsequent to viral infections. Strong correlations between elevated body mass index (BMI), type-2-diabetes and increased risk of hospitalization after pandemic influenza H1N1 infection have been described. Similarly, a correlation between elevated blood glucose level and SARS-CoV-2 infection severity and mortality has been described, indicating MetS as an important predictor of clinical outcomes in patients with COVID-19. Adipose secretome, including two of the most abundant and well-studied adipokines, leptin and interleukin-6, is involved in the regulation of energy metabolism and obesity-related low-grade inflammation. Similarly, skeletal muscle hormones—called myokines—released in response to physical exercise affect both metabolic homeostasis and immune system function. Of note, several circulating hormones originate from both adipose tissue and skeletal muscle and display different functions, depending on the metabolic context. This review aims to summarize recent data in the field of exercise immunology, investigating the acute and chronic effects of exercise on myokines release and immune system function.

## 1. Introduction

Metabolic Syndrome (MetS) is a cluster of metabolic alterations mostly related to visceral adiposity and includes arterial hypertension, hyperglycemia and dyslipidemia [1]. The prevalence of obesity is increasing worldwide, contributing to the increasing incidence of cardio-metabolic disorders [2]. In accordance with evidence supporting a strong association between visceral adiposity and metabolic alterations, MetS is characterized by a chronic systemic inflammatory state, which, in turn, is associated with a greater risk of type 2 diabetes mellitus (T2DM) and cardiovascular disease. Importantly, MetS increases the risk of severe disease due to viral infections [1].

In general, excess adiposity results when energy intake exceeds energy expenditure, due to both overnutrition and insufficient physical activity, leading to a positive energy balance. There is accumulating evidence that the dysfunctional adipocyte is a major source of pro-inflammatory cytokines, promoting systemic low-grade inflammation and thus further contributing to the development of MetS and obesity-related disorders [3]. In particular, pro-inflammatory immune cell infiltration in terms of macrophages, effector and memory T-cells, T regulatory cells, natural killer and natural killer T cells and granulocytes are a major contributor to an inflammatory milieu in obesity [4]. This immune system activation might represent the link between MetS and the increased risk of severe disease due to viral infection.

Exercise is considered a form of physical stress, acting as an immune system modulator, through both neuro-endocrine and metabolic adaptations, underlying muscle contraction [5]. In the early 20th century, Larrabee, Tileston and Emerson were the first to observe leukocytosis among athletes participating in the Boston Marathon two hours after the end of the race [6]. In accordance with their observations, several studies indicated that prolonged and intensive exercise was associated with transient immune dysfunction, elevated inflammatory biomarkers and increased risk of upper respiratory tract infections [7,8,9,10,11,12].

In the last decades, advances in mass spectrometry and genetic testing allowed for a better understanding of the emerging correlations between “homic” sciences (metabolomics, proteomics, lipidomics, genomics), intestinal microbiota, physical exercise and the immune system [13,14]. Like adipose tissue, skeletal muscle tissue is also considered a veritable endocrine organ, releasing a number of circulating chemical mediators called “myokines”, which are involved in the regulation of metabolic and immune health [15,16,17].

Myokines specifically act at the adipose tissue level, by reducing and regulating fat mass expansion, favoring the beige/brown phenotype which, in turn, improves metabolic homeostasis. At a systemic level, myokines improve insulin sensitivity in target tissues and organs (e.g., liver, muscles) [18]. The endocrine adaptation of skeletal muscle in response to exercise training can therefore result in systemic anti-inflammatory effects, counteracting deleterious consequences of metabolic syndrome on the immune system [19].

As recommended by current guidelines, both physical exercise and adequate nutrition are able to modulate myokines release, thus potentially contributing to improve insulin sensitivity and mitigate the risk associated with cardiometabolic diseases [20]. The aim of this narrative review is to report and discuss the latest evidence on the impact of physical exercise and nutritional status on mediators of inflammation. In particular, the review summarizes how metabolic dysregulation impairs immune responses during influenza virus and coronavirus infection with a particular focus on patients with MetS at an increased cardio-metabolic risk. The novelty of this manuscript resides in the change of perspective adopted to discuss the impact of metabolic homeostasis on immune function in response to physical exercise. Fat accumulation is widely considered the most important actor, releasing hormones able to induce chronic inflammation with immune cells infiltration, as well as metabolic alterations. On the other hand, skeletal muscle represents a veritable endocrine organ, secreting a number of contraction-induced hormones, which are known to regulate immunometabolism. To illustrate the complex relationships between physical exercise, skeletal muscle, adipose tissue and immune system, a brief description of the main hormones secreted by skeletal muscle and their physiological significance will be also provided.

## 2. Metabolic Syndrome and Viral Infections

Visceral adiposity and glucose intolerance subsequent to overfeeding, as well as to altered insulin signaling and progressive loss of beta-cell function, represent well established features of MetS and type 2 diabetes (T2DM) [1,21,22]. Patients with obesity and T2DM are particularly vulnerable to viral infections, although the underlying mechanisms are not well established [23,24,25,26].

A 2009 clinical study by Louie et al., investigating the impact of BMI on the incidence of severe influenza infection among California residents hospitalized with H1N1, reported that half of the 534 patients infected by influenza virus were obese [27]. In accordance with their findings, another study revealed that adults with a body mass index (BMI) > 30 kg/m^2^ had a 1.45 to 3 times increased risk for hospitalization during influenza seasons in Canada [28]. Similarly, patients with T2DM demonstrated increased influenza-related mortality, showing an increased risk of hospitalization and intensive care unit (ICU) admission upon hospitalization [29]. The greater prevalence of T2DM in individuals infected by H1N1 developing fatal disease complications is related to chronic hyperglycemia, which impairs innate and humoral immune systems, with reduced function of T cells and neutrophils (Table 1) [30]. Similarly, obese individuals show alterations at different steps of the innate and adaptive immune response, characterized by a state of chronic and low-grade inflammation, which seems to be a major determinant in the severity of viral infections in obesity. Indeed, the visceral adipose tissue secretome releasing leptin and other pro-inflammatory cytokines is able to negatively affect immune system function, for example, by reducing macrophage activation after antigen presentation [31,32]. This may explain the increased susceptibility and delayed recovery of viral infections in obese/diabetic individuals. Importantly, macrophage dysfunction may alter vaccine efficacy together with virus pathogenicity [33,34].

Recently, several studies discussed the impact of MetS on COVID-19 clinical outcomes, showing that elevated BMI, as well as obesity-related metabolic dysfunctions, including T2DM, represent important risk factors for complications and mortality following SARS-CoV-2 infection [35,36,37]. In particular, hyperglycemia has been indicated as a predictor for the fatality of COVID-19 infection (Table 1) [38,39,40].

Elevation of initial blood glucose levels was identified as an independent risk factor for in-hospital mortality among critical cases [38]. Moreover, patients with new-onset hyperglycemia, even in the absence of diabetes diagnosis, showed poorer outcome compared to the normoglycemic individuals, as well as those with pre-existing diabetes [39]. On the other hand, obesity per se induces detrimental effects in patients with SARS-CoV-2 infection, potentially leading to worse outcomes, including respiratory and multiple organ failure and thus increasing the mortality risk [41]. Several factors are implicated in the association between obesity and increased risk of hospitalization due to a COVID-19 infection. First, dysfunctional visceral fat secretome, including pro-inflammatory cytokines and adipokines, impairs immune function, through a direct regulation of both innate and adaptive immune response, as mentioned above [42]. Second, abdominal fat accumulation hampers diaphragmatic movement, reducing basal lung expansion during inspiration. For this reason, obese individuals are more likely to have worse lung function and respiratory symptoms than individuals with a normal BMI [43,44].

Given the increasing prevalence of obesity worldwide in recent years, these data highlight that increased risk of respiratory infections—with the relative burden of severe complications—exists in obese subjects.

Despite such evidence, the impact of obesity and metabolic alterations on viral infections severity is underestimated probably due to the fact that BMI is generally not identified as a significant problem in primary care. For this reason, it is usually not inserted in medical records upon hospitalization, unless the patient undergoes invasive surgery [45]. This could represent bias in the context of retrospective clinical studies focusing on the impact of obesity on viral infection outcomes. Other bias, such as those correlating pneumonia to obesity, may be due to the difficulty in reading X-ray reports in overweight people or the fact that early studies in this field have used variable BMI values to define obesity [46]. Obese individuals were shown to have a higher morbidity and mortality from COVID-19. A major concern is that vaccines could be less efficient in subjects with obesity [47].

## 3. Adipose Tissue Dysfunction and Immunomodulation in MetS

The histological and functional complexity of the adipose organ goes far beyond its apparent nature as a ‘fat reservoir’ [48]. Adipose tissue is a specialized form of connective tissue encompassing vascular stroma (containing endothelial cells, pericytes, fibroblasts and pluripotent stem cells) and mature adipocytes, able to store fats in the form of triglycerides [49]. Adipose tissue is mainly located under the skin at the level of the hypodermis and internally, in the peritoneal area, at the level of the large omentum, behind the intestine, around the kidneys and the pericardial [50].

At the histological-functional level, we distinguish mainly between two types of adipose tissue. White adipose tissue (WAT), located both in the hypodermis and viscerally, is characterized by adipocytes containing a single large lipid droplet and low mitochondrial density with low metabolic activity [51]. Brown adipose tissue (BAT), which is more prevalent in infants than in adults, is characterized by adipocytes with a number of small lipid droplets, higher mitochondrial density and with important thermogenic function [52,53]. Recently, beige adipose tissue with intermediate characteristics of both tissues has been identified. The onset of the beige adipocyte is still controversial, but it is well established that it is capable of expressing discrete quantities of thermogenin in response to both physical exercise and calorie restriction [54]. Genetic, hormonal and environmental factors are involved in beige/BAT development and metabolism as well as in the browning process of WAT [55,56]. Adipose depots are considered a veritable endocrine organ secreting different adipokines which regulate several neuroendocrine functions, including energy expenditure, insulin sensitivity, lipid and glucose metabolism, endothelial function, blood pressure and immunity [57,58]. Visceral fat also releases free fatty acids (FFAs) through lipolysis into the portal bloodstream, determining intrahepatic fat accumulation and skeletal muscle insulin resistance [59].

In obesity, dysfunctional adipose tissue promotes systemic low-grade inflammation which, in turn, contributes to the development of obesity-related diseases [3]. In particular, adipose tissue hypertrophy and hyperplasia determine local hypoxia and cell death, as well as extracellular matrix remodeling and mitochondrial dysfunction, leading to immune cell infiltration [60]. Adipocytes secrete chemoattractant factors, including the stromal cell-derived factor-1 α (SDF-1α), which promotes recruitment and survival of cluster of differentiation 4+ (CD4+) T-lymphocytes, resulting in adipose tissue inflammation and subsequent recruitment of proinflammatory macrophages [61,62].

On the other hand, macrophages infiltrating adipose tissue shift to a pro-inflammatory phenotype, forming typical crown-like structures around adipocytes and secreting pro-inflammatory cytokines [63]. Of note, the most studied adipokine leptin plays an essential role in regulating immune system function, promoting an obesity-related low-grade inflammation state. Preclinical studies using transgenic mice clarified the role of leptin as a potent immune modulator [64,65].

Accordingly, genetic mutations in leptin or the leptin receptor promote fat accumulation and obesity development with immune system dysfunction in humans [66]. With regard to the adaptive immunity response, pro-inflammatory effects of leptin have been described. Leptin receptors are expressed in CD4+ T-lymphocytes [67], where leptin signaling regulates survival, proliferation, cytokine release and differentiation [68,69]; on the other hand, the proliferation of regulatory T cells (Treg cells), showing anti-inflammatory activity, is inhibited by leptin in human cells ex vivo [70]. Similarly, leptin signaling mediates pro-inflammatory effects in innate immune cells, including macrophages where the adipokine induces both phagocytosis and cytokine production [71,72].

Adipose tissue and the immune system share cytokines secretion whose role can vary enormously depending on their targets, as well as the metabolic context in which they are secreted [73]. Given the strong correlation between obesity, low-grade inflammation and metabolic diseases, inflammation pathways represent a valid target for the treatment of obesity-related comorbidities [74].

## 4. Skeletal Muscle and Exercise Immunology

The skeletal muscle system is essential to maintain general health and well-being and plays a pivotal role in ensuring vital functions such as movement, postural support, breathing and thermogenesis. Sarcopenia is a clinical condition characterized by progressive decline in skeletal muscle mass and strength occurring in aging, and it is considered a predictor factor of fractures, disability and functional impairments associated with significant morbidity and mortality. Interestingly, Sarcopenia is also associated with increased infection susceptibility, thus representing a potential link between impaired muscle function and impaired immune response. On the other hand, sarcopenic obesity occurs when reduced skeletal muscle mass and strength are accompanied by visceral fat accumulation, which induces lipotoxicity as well as systemic inflammation, as already discussed [75,76,77,78,79,80,81]. The combination of muscle loss and ectopic lipid deposition, induced by visceral fat expansion, leads to impaired immune response to a greater extent compared to sarcopenia or obesity alone and [82] reduces the ability to respond to metabolic stress [83]. Such evidence is of particular interest due to the current global pandemic caused by COVID-19.

Exercise immunology is an important research area investigating acute and chronic effects of exercise on the immune system function, including clinical benefits of the exercise–immunity relationship and nutritional influences on the immune response to exercise. Skeletal muscle contraction induced by physical exercise triggers neuro-endocrine and metabolic adaptations, which potentially influence the immune system function [5]. Like adipose tissue, skeletal muscle tissue displays endocrine properties [84] essential in regulating physiological response to physical exercise, through the release of chemical mediators, the so-called myokines, capable of exerting effects at an autocrine, paracrine and even endocrine level through the blood circulation on different target tissues (Figure 1) [85,86].

Notably, in 2013, Raschke and Eckel discussed the existence of adipo-myokines. Indeed, several myokines, whose expression is regulated by physical exercise-induced contraction, are also secreted by adipocytes thus playing a dual role as mediators of both inflammation and metabolic health, depending on the circumstances [87]. This paragraph focuses both on the main myokines, produced by muscle tissue in response to physical exercise and involved in metabolic homeostasis and immune system modulation, as well as on adipo-myokines, capable of exerting bioactive functions that vary depending on the source tissue and the metabolic context [85].

## 5. Myokines

**Fibroblast growth factor 21** (FGF21) is an endocrine hormone secreted by several organs and involved in energy homeostasis regulation. In particular, FGF21 is induced in response to mitochondrial dysfunction, starvation and endoplasmic reticulum stress, and it is considered a powerful endocrine mediator, capable of improving glucose tolerance, lipid metabolism and energy expenditure in murine models [88,89].

Some studies suggest that skeletal muscle FGF21 favors browning by regulating the expression of the mitochondrial protein uncoupling protein 1 (UCP1) in white adipose tissue depots [90] Despite the aforementioned studies, the role of FGF21 as an exercise-induced myokine remains controversial.

A clinical study demonstrated that physiological insulin concentrations induce muscle expression of FGF21, which, in turn, correlates with increased hormone circulating levels in overweight/obese diabetic individuals. Interestingly, physical exercise showed no effects on FGF21 expression in skeletal muscle and adipose tissue, although a significant improvement in insulin sensitivity was observed in these subjects, suggesting that FGF21 is a direct target of insulin action, even in the presence of insulin resistance [91]. In addition to that, several studies showed that aerobic exercise increases in FGF21 serum levels in healthy individuals, immediately after exercise session and after two weeks of training [92,93]. Further studies are necessary to clarify the therapeutic potential of muscle FGF21.

**Myogenin** is a transcription factor essential for myogenesis and skeletal muscle repair [94]. Different stimuli, including muscle trauma induced by physical training, are able to promote satellite cells (SCs) proliferation, thus increasing the expression of myogenic markers [95]. Myogenin is one of the so-called “regulatory myogenic factors” (along with myogenic factor 5 (Myf5) and myoblast determination protein 1 (MyoD)) that plays a key role in the transition from satellite cells to myoblasts and their fusion to restore damaged myofibrils [96]. Accordingly, myogenin gene expression increases in SCs isolated from human skeletal muscle biopsy samples following eccentric contractions [97]. On the other hand, recruited immune cells play a critical role in muscle repair.

During muscle regeneration, SCs are reprogrammed by infiltrating neutrophils and monocytes, through pro-inflammatory cytokines release, which represents the initial stages of myogenesis. Then, activated macrophages acquire anti-inflammatory properties to resolve inflammation and facilitate tissue recovery, determining the transition from the proliferative stage to the differentiation and growth stage of myogenesis [98,99,100].

**Myonectin**, also known as CTRP15, is a very important exercise-training-responsive myokine, regulating lipid metabolism in skeletal muscle, adipose tissue and liver and showing a significant positive impact on NEFA (non-esterified fatty acids) levels, in mice [101]. Of note, lipid oxidation represents the most effective molecular adaptation to training, and it is accompanied by an increase in mitochondria number and activity in response to physical exercise [102,103].

Increased myonectin circulating levels correlate with insulin resistance in humans. On the other hand, the same study indicates that obese subjects show reduced myokine levels, demonstrating an inverse relationship between myonectin and BMI values [104], thus suggesting that increased myonectin levels in the presence of IR may represent a compensatory mechanism. In accordance with this evidence, a recent study showed that a moderate-intensity aerobic exercise program (between 50% and 70% of maximum heart rate) is capable not only of increasing myonectin levels but also of reducing insulin resistance in obese subjects [105]. Given the described role of myonectin in energy metabolism regulation, the specific impact of this myokine on immune system in response to exercise needs to be explored.

**Brain-derived neurotrophic factor (BDNF)** is a small protein mainly produced by the central nervous system (but also by skeletal muscle) and plays an important role in neurogenesis processes related to memory and learning. Studies conducted so far show that physical exercise is able to positively stimulate BDNF release [106,107]. In particular, the increase in BDNF levels after aerobic exercise is closely correlated to energy expenditure, therefore, high intensity and/or high-volume training is more effective in inducing an increase in BDNF blood concentration. [106].

BDNF displays both autocrine action at a muscular level in muscle regeneration, as well as in muscular adaptation to exercise, and endocrine action at a central nervous system (CNS) level, as revealed by a positive neurotrophic impact in terms of enhanced cognitive function correlated to increased plasma levels of BDNF after physical exercise, especially in elderly subjects [108]. Moreover, BDNF exerts anorexic actions at a CNS level by interacting with hypothalamic centers regulating appetite [109]. Interestingly, it has been demonstrated that, like other neurotrophins, BDNF contributes to the functioning of both innate and acquired immunity [110]. Such immunomodulatory property results in potential neuroprotective effects in the presence of neurodegenerative diseases [111].

As regards metabolic aspects, BDNF regulates both glucose and lipid metabolism [112]. In particular, physical exercise enhances circulating levels of BDNF, which are decreased in obesity and type 2 diabetes mellitus, improving lipid oxidation and insulin sensitivity [113]. Of note, BDNF preserves pancreatic beta-cell function in experimental models [114]. Furthermore, emerging evidence indicates a link between BNDF and COVID-19 disease. SARS-CoV-2 infection dampens BDNF synthesis and release, thus favoring COVID-19 associated neurologic symptoms. For this reason, BDNF levels have been proposed as a predictive factor of intensive care unit admission in COVID-19 patients [115].

**Monocyte chemoattractant protein-1 (MCP-1/CCL2)** is the best-characterized chemokine released by skeletal muscle in response to injury. It is involved both in the recruitment of monocytes and T lymphocytes, thus triggering the beneficial inflammation essential to support muscle regeneration but also in the alteration of insulin sensitivity and obesity-related low-grade inflammation [116,117].

Indeed, diabetic patients show elevated MCP-1 circulating levels, and it has been shown that exercise is able to significantly reduce MCP-1 plasma levels [118,119]. Notably, even mild physical exercise (10,000 steps a day, 3 times a week, for 8 weeks) is capable of downregulating MCP-1 expression in sedentary subjects, supporting the beneficial anti-inflammatory benefit of low-intensity exercise [120]. Although there is a lack of reliable data on human models, the possible role of MCP-1 as a connecting link between exercise, immune response and skeletal muscle insulin resistance needs to be established by further studies.

## 6. Adipo-Myokines

**Interleukin 6 (IL-6)** is a protein composed of 212 amino acids, produced from adipose tissue, skeletal muscle and immune system cells [121]. It is well established that IL-6 plays a different role (inflammatory or anti-inflammatory) depending on the tissue where it is expressed, as well as on its mechanism of action. Despite the well-described pro-inflammatory effects of systemic IL-6 in metabolic diseases, clinical studies demonstrate that IL-6 secreted by skeletal muscle in response to physical exercise exerts anti-inflammatory effects with beneficial metabolic responses [8]. Indeed, IL-6 membrane receptor (IL-6R) requires a pair of glycoproteins (gp130) as co-receptors to form the IL-6–IL6R–gp130 complex in order to activate the intracellular signaling transduction. However, a small number of cells express IL-6R, whereas gp130 is present in almost all cell populations; thus, in the absence of IL-6R, IL-6 requires a soluble receptor (sIL-6R) to initiate signaling [122]. When IL-6 exerts its effects via soluble receptors, the signaling pathway is called “trans” and has a pro-inflammatory effect; on the other hand, when membrane receptors are expressed by target cells, the triggered pathway is called “classic” and displays anti-inflammatory effects. Plasma levels of IL-6 increase as a result of muscle exercise and depend closely on the intensity and duration of the stimulus [123]. During physical exercise, skeletal muscle secretes IL-6, whereas sIL-6R is not expressed [124]. In this tissue, sIL-6R is produced as a result of enzymatic reactions in the presence of other chemical mediators of inflammation, such as tumor necrosis factor alpha (TNF-alpha) produced by dysfunctional adipocytes. Skeletal muscle IL-6 specifically acts at the level of the target cells expressing IL-6R, inducing a cascade of signals that culminate in biological actions aimed at improving the body’s metabolic profile in terms of energy substrates, improved insulin sensitivity and fatty acids oxidation, increased lipolysis and glycogenolysis in liver [125]. In addition, skeletal muscle IL-6 plays an important role in processes related to muscle growth, stimulating satellite cells, the differentiation, browning of fat deposits and cardiovascular protection [126].

**Interleukin 15 (IL-15)** is a cytokine discovered in 1994 in T lymphocytes and generally associated with inflammatory processes; IL-15 is also produced by skeletal muscle in response to physical exercise [127], and its receptors (interleukin 15 receptor alpha (IL-15Ralfa)) are expressed by different tissues involved in metabolic homeostasis [128].

Very similar signaling has been observed for IL-15 and IL-6, with increased glucose transporter type 4 (GLUT-4) expression and tanslocation, thus resulting in improved glucose uptake and energy metabolism [129,130]. At the adipocyte level, IL-15 promotes lipolysis and stimulates mitochondrial activity in brown adipocytes, increasing the expression of UCP1 [131], as well as regulating cell proliferation by inhibiting pre-adipocytes. Such a property is of particular interest due to the well-known role of adipose tissue expansion in the increase in mechanical stress, hypoxia and low-grade inflammation [132]. Short sessions of aerobic activity seem to increase IL-15 levels in both lean and overweight/obese subjects [133]. A more recent study showed an increase (2.2-fold) in serum IL-15 levels even after an acute trail run session (35 km); in the same study, a greater increase (13.2-fold) in circulating IL-6 levels was observed [127].

**Leukaemia inhibitory factor (LIF)** is a small protein composed of 181 amino acids. It plays an important role in hematopoietic processes related to the differentiation of myeloid line cells (monocytes, neutrophil granulocytes, eosinophils, basophils, erythrocytes, megakaryocytes, dendritic cells) [134].

It shares the same gp130 co-receptors with IL-6, thus performing an anti-inflammatory function. A recent preclinical study hypothesized a role for LIF in improving glucose uptake, at muscle level, as well as in inhibiting adipocyte proliferation [135]. In humans it has been shown that both aerobic activity and exercise with overloads regulate LIF mRNA expression in skeletal muscle [136].

**Irisin (FNDC5)** has been identified for the first time in 2012 by Spiegelman’s group. It is a hormone produced by muscle tissue following physical exercise, and it plays an essential role in the cross-talk between different tissues/organs, including adipose tissue, bone, brain and skeletal muscle [137]. Irisin promotes browning of adipose tissue with beneficial effects in term of weight loss, thermogenesis activation and improved glucose tolerance. Moreover, it stimulates new bone formation and displays neuroprotective effects, through the upregulation of BDNF, thereby also enhancing cognitive capacity, by increasing the number of synapses [138,139]. As regards metabolic action, reduced irisin circulating levels have been associated with angiopathy and renal dysfunction development in T2D mellitus [140].

**Myostatin (MSTN)** is a protein discovered in 1997 by McPherron and Se-Jin Lee with an important role in regulating skeletal muscle growth [141]. Its action is in fact regulated by the presence of follistatin, which is produced by the liver in response to physical activity and is known to antagonize MSTN, promoting browning both in vitro and in vivo [142]. Interestingly, MSTN is also expressed in the adipose tissue exerting a dual function, either inhibiting or promoting adipogenesis depending on the context. Despite many aspects that need to be clarified, preclinical studies showed that MSTN genetic deletion leads to increased muscle mass, with reduced fat mass and resistance to diet-induced obesity [143,144].

Clinical studies display a direct effect of physical exercise on skeletal muscle MSTN mRNA expression, which is reduced after sessions of variable duration of exercise, especially if the exercise is structured on a multiweekly program aimed at sedentary subjects [145,146].

**Meteorin-like protein precursor (METRNL)** is a factor induced in muscle in response to exercise and in adipose tissue as a result of exposure to cold [147]. Its biological activity is mainly expressed in increasing energy expenditure, improving glucose tolerance and inducing thermogenesis in beige adipose tissue, as well as in promoting anti-inflammatory cytokines release [148,149]. Despite the established role of METRNL in thermogenesis, a clinical study demonstrated that aerobic activity performed at low temperatures does not improve its expression. Moreover, a positive association has been found between METRNL and Irisin, whose circulating levels are elevated in individuals with T2D [150]. Further studies are necessary to clarify the metabolic impact of this adipo-myokine.

**Beta-Aminoisobutyric Acid (BAIBA)** is a recently described adipo-myokine, whose importance has increased due to its implications in metabolic homeostasis [151]. In particular, two enantiomers exist in biological systems: R-BAIBA and S-BAIBA. BAIBA release is induced, at the skeletal muscle level, as a result of physical exercise, playing a protective role against diet-induced obesity due to its ability to induce browning of WAT in animal models. This metabolic effect is also related to increased fatty acid oxidation and ketone body production, as well as increased expression of carnitine palmitoyltransferase 1 (CPT-1) in the liver, resulting in improved insulin sensitivity [152]. It has been shown that BAIBA release protects osteocytes from apoptosis, thus preventing bone tissue dysfunction by reducing reactive oxygen species (ROS) production at the mitochondrial level [153]. Although many mechanisms related to its biological action still need to be clarified, it has been hypothesized that BAIBA exerts its action on lipid metabolism and insulin sensitivity by restoring and enhancing the biological action of leptin [154]. Of note, acute aerobic exercise sessions induce increase in both the enantiomer R (13%) and S (20%) in the plasma of healthy subjects [155]. Accordingly, a general BAIBA increase (17%) was observed in healthy sedentary subjects after aerobic exercise programs performed three times per week over a period of 20 weeks [156].

Overall, it seems evident that physical exercise is able to modulate systemic inflammation through adipo-myokines release, thus potentially inducing different immune changes depending also on exercise intensity. Thus, physical exercise together with adequate nutrition may be the main preventive measures to maintain metabolic health and energy balance. Both adipose tissue and skeletal muscle are considered endocrine organs secreting hormones able to modulate metabolic homeostasis, as well as immune system function. Physical-exercise-induced myokines and muscle-adipose tissue cross-talk have important implications for health and metabolic diseases. Generally, adipokines play a pivotal role in low-grade inflammation associated with visceral fat accumulation, whereas myokines are released in response to muscle contraction and show anti-inflammatory and beneficial metabolic effects. The immune system is responsive to the physiological stress induced by skeletal muscle contraction. Exercise immunology takes advantage from this observation and suggests that physical exercise may induce beneficial metabolic changes in immune cells, through myokines secretion, by regulating cellular energy sensors. On the other hand, it is well established that physical inactivity contributes to obesity and MetS development [157]. Actually, evidence supporting the correlation between physical inactivity, MetS and infectious risk is limited due to the lack of epidemiological studies, as well as to social influences considering physical exercise as a hobby more than preventive/therapeutical approach aimed to maintain individual health. Yet it should be considered that physical activity promotion is at the heart of the Global Action Plan on Physical Activity 2018–2030 proposed by the World Health Organization (WHO). Such a plan involves several countries and aims to improve social, economic, political and infrastructural processes able to counteract physical inactivity by promoting a culture of movement. To date, the rate of physical inactivity in countries with strong industrialization and urban development, reaches almost 70%; the WHO aims to reduce this value by at least 15%, by 2030, in all age groups in order to promote the concept of active society. The achievement of this goal will take into account differences in motor possibilities between population groups from each country involved. It is important to highlight that there are still deeply rooted beliefs, already discussed in this manuscript, indicating that a certain amount of exercise can exert opposite effects, not only worsening the inflammatory status but also dysregulating the immune response and exposing the subject to a specific type of infectious risk [158].

## 7. Conclusions

Obesity correlates with altered metabolic homeostasis mostly due to visceral fat accumulation, which, in turn, promotes systemic low-grade inflammation, affecting the immune response to infections. Thus, MetS represents a risk factor not only for non-communicable diseases but also in the assessment of infectious risk. In particular, obese patients display respiratory dysfunction due to altered respiratory mechanisms, caused also by mechanical compression of the diaphragm and reduced lung volume. Such impairment in the respiratory function due to visceral fat accumulation can facilitate pneumonia infections in these individuals, increasing the risk of pulmonary hypertension and cardiac stress. Obesity-related comorbidities increase the risk of organ failure associated with pneumonia [159]. Such evidence is of particular interest with regard to pandemic COVID-19. Indeed, glucose homeostasis, lipid profile as well as blood pressure control are essential to determine the clinical outcome in these patients [160]. In this context, the assessment of anthropometric and metabolic parameters, including BMI, waist and hip circumferences and glucose and insulin levels, is crucial to estimate the risk of complications in patients with COVID-19. In particular, insulin resistance represents a strong determinant of metabolic health impairment, cardiac dysfunction and cardiovascular disease (CVD)-related mortality. Future studies will improve knowledge in the field of personalized nutrition, also providing physical exercise guidelines based on real needs of the population, thus taking into account social, economic and environmental factors influencing people’s lifestyle. This approach may be essential to improve the quality of life, as well as to prevent the development of cardiometabolic disease, showing negative impact on infections outcome and rehabilitation programs, thus reducing individual life expectancy.

## Figures and Tables

**Figure 1 nutrients-14-02054-f001:**
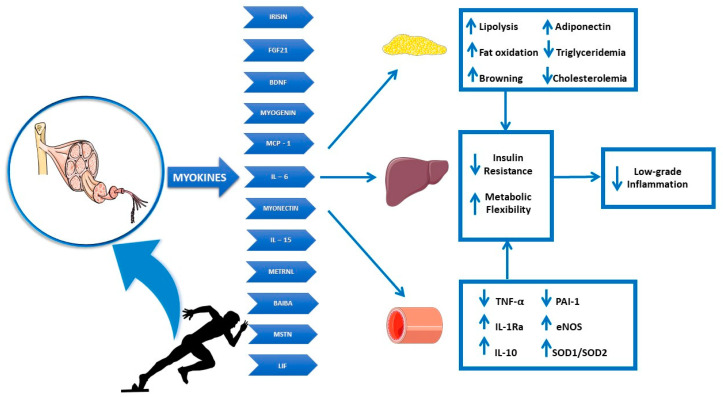
Skeletal muscle as an endocrine organ: metabolic effects of muscle-contraction-induced myokines. Physical exercise stimulates myokine release showing endocrine function at target tissues level, including adipose organ, liver and endothelium. Indeed, the skeletal muscle secretome plays an essential role in the maintenance of whole-body metabolic homeostasis, regulating lipolysis, fat oxidation, inflammation and insulin sensitivity. In particular, myokines release induced by physical exercise contributes to the immune system stimulation, producing an anti-inflammatory cellular response with potential protective effects against infections. Abbreviations: TNF-α (Tumor necrosis factor-alpha), IL-1ra (Interleukin-1 Receptor Antagonist), IL-10 (Interleukin-10), PAI-1 (Plasminogen activator inhibitor-1), eNOS (Endothelial nitric oxide synthase), SOD1 (Superoxide Dismutase 1), SOD2 (Superoxide Dismutase 2), FGF21 (Fibroblast growth factor 21), BDNF (Brain-derived neurotrophic factor), MCP-1 (Monocyte chemoattractant protein-1), IL-6 (Interleukin-6), IL-15 (Interleukin-15), METRNL (Meteorin-like protein), BAIBA (Beta-Aminoisobutyric acid), MSTN (Myostatin), LIF (Leukemia inhibitory factor), (↓) decrease, (↑) increase.

**Table 1 nutrients-14-02054-t001:** Evidence of increased severity of H1N1 influenza and COVID-19 in diabetes and obesity based on clinical studies.

Metabolic Alterations and Viral Infection Outcomes				
First Author	Viral Infection	Results	Type of Publication	Country
Louie, J.K., 2011 [27]	H1N1 Influenza	Half of 534 adult case patients hospitalized with 2009 H1N1 infection were obese. Extreme obesity (BMI ≥ 40 kg/m^2^) was associated with increased odds of death, thus representing an independent risk factor for mortality.	Article	California, USA
Kwong, J.C., 2011 [28]	H1N1 Influenza	Logistic regression to examine the association between BMI and hospitalization for selected respiratory diseases in a cohort of 82,545 adults over 12 influenza seasons (1996–1997 through 2007–2008) indicates that severely obese individuals (Class II or III, BMI > 35 kg/m^2^) with and without chronic conditions are at increased risk for respiratory hospitalizations during influenza seasons.	Article	Canada
Allard, R., 2010 [29]	H1N1 Influenza	Diabetes triples the risk of hospitalization after influenza A (H1N1) p and quadruples the risk of ICU admission once hospitalized.	Article	Canada
Pranata, R., 2021 [36]	SARS-CoV-2	A total of 34,390 patients from 12 studies were included in this meta-analysis. Increased BMI was associated with increased poor outcome in patients with COVID-19.	Meta-Analysis	Several countries
Guo, W., 2020 [37]	SARS-CoV-2	COVID-19 patients with diabetes (*n* = 24) were at higher risk of severe pneumonia, release of tissue injury-related enzymes, excessive uncontrolled inflammation responses and hypercoagulable state associated with dysregulation of glucose metabolism. Diabetes should be considered as a risk factor for a rapid progression and bad prognosis of COVID-19.	Article	China
Wu, J., 2020 [38]	SARS-CoV-2	Elevation of admission blood glucose level was an independent risk factor for progression to critical cases/death among non-critical cases in a cohort of 2041 consecutive hospitalized patients with COVID-19.	Article	China

Abbreviations: BMI (body mass index), ICU (intensive care unit).

## Data Availability

Not applicable.

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
