# Peer review of "Influence of Nutritional Status and Physical Exercise on Immune Response in Metabolic Syndrome"

_nutrients, 2022, doi:10.3390/nu14102054_

Round 1
Reviewer 1 Report
This is an extremely extrnsive review. I would recommend adding more information into the Abstract section.
Author Response
Dear Editors and Reviewers,
first, we would like to thank you for the valuable impulses that allowed us to improve the quality of the manuscript. All changes made are highlighted by yellow color, in the revised version of the manuscript, to facilitate the review process.
Hoping that we have satisfied your requests as much as possible, we kindly ask you to re-evaluate our paper.
The Authors
—
REVIEWER N.1
This is an extremely extensive review. I would recommend adding more information into the Abstract section
We thank the Reviewer for the useful comment. We added few sentences in the abstract to clarify the aim of the review and the main findings discussed into the manuscript.
Reviewer 2 Report
Please discuss the novelty of this manuscript.
- Title must be revised.
- Please clarified the goal of this manuscript
Author Response
Dear Editors and Reviewers,
first, we would like to thank you for the valuable impulses that allowed us to improve the quality of the manuscript. All changes made are highlighted by yellow color, in the revised version of the manuscript, to facilitate the review process.
Hoping that we have satisfied your requests as much as possible, we kindly ask you to re-evaluate our paper.
The Authors
REVIEWER N.2
- - Title must be revised.
We agree with the Reviewer. In the revised version of the manuscript we changed the title as follows: “Influence of nutritional status and physical exercise on immunity response in metabolic syndrome”.
- - Please clarified the goal of this manuscript
- - Please discuss the novelty of this manuscript.
We revised the manuscript according to the suggestions of the Reviewer.. In order to better clarify the objective and novelty of the review, we added two paragraphs both in Introduction and Conclusions sections.
Reviewer 3 Report
In the above-mentioned review article, the Authors aimed to summarize recent data in the field of exercise immunology, investigating the acute and chronic effects of exercise on myokines release and immune system function. The article is written clearly and only some minor corrections are suggested:
- English editing for some minor corrections (i.e. typographical errors) is recommended.
- Several abbreviations in the text should be explained, as well as the abbreviations below Figure 1.
- Page 2 and Table 1-the units for BMI are missing (i.e. kg/m2).
Author Response
Dear Editors and Reviewers,
first, we would like to thank you for the valuable impulses that allowed us to improve the quality of the manuscript. All changes made are highlighted by yellow color, in the revised version of the manuscript, to facilitate the review process.
Hoping that we have satisfied your requests as much as possible, we kindly ask you to re-evaluate our paper.
The Authors
REVIEWER N.3
In the above-mentioned review article, the Authors aimed to summarize recent data in the field of exercise immunology, investigating the acute and chronic effects of exercise on myokines release and immune system function. The article is written clearly and only some minor corrections are suggested:
We deeply appreciate the Reviewer's positive feedback.
- English editing for some minor corrections (i.e. typographical errors) is recommended.
- Several abbreviations in the text should be explained, as well as the abbreviations below Figure 1.
- Page 2 and Table 1-the units for BMI are missing (i.e. kg/m2).
We thank the Reviewer for his comments. We revised the text by fixing spelling mistakes, abbreviations, and captions under tables and figures.
Round 2
Reviewer 2 Report
The authors have done their best to address all of my comments and responded accordingly and appropriately. Thus, the manuscript is greatly improved from the original version.
Author Response
Many thanks to the reviewer for the valuable help in improving the paper.